# Crop Yield and Nutrient Efficiency under Organic Manure Substitution Fertilizer in a Double Cropping System: A 6-Year Field Experiment on an Anthrosol

**Yan Han [†], Fenglian Lv [†], Xiaoding Lin, Caiyun Zhang, Benhua Sun, Xueyun Yang * and Shulan Zhang ***

Key Laboratory of Plant Nutrition and the Agri-Environment in Northwest China, Ministry of Agriculture and Rural Affairs/College of Natural Resources and Environment, Northwest A&F University, Xianyang 712100, China

\* Correspondence: xueyunyang@nwafu.edu.cn (X.Y.); zhangshulan@nwafu.edu.cn (S.Z.)

† These authors contributed equally to this work.

**Abstract:** The combination of organic manure and inorganic fertilizer plays a role in increasing crop yield and nutrient efficiency, but such effectiveness varies with crop, soil, management, and climate. Here, a 6-Year field experiment was conducted to evaluate the effects of substituting organic manure with inorganic fertilizer on crop yield, grain protein content, and nitrogen and phosphorus efficiency under a winter wheat-summer maize cropping system on Anthrosol. Five treatments were included: recommended nitrogen (N), phosphorus (P) and potassium (K) fertilizers (NPK), 75% NPK + 25% organic manure (M), 50% NPK + 50% M, 25% NPK + 75% M, and 100% M, respectively. Wheat, maize, and annual yield were 1643–8438 kg ha$^{-1}$, 4847–11,104 kg ha$^{-1}$, and 10,007–17,496 kg ha$^{-1}$. Organic manure treatments produced the same crop yield as NPK treatment except for 100% M. Grain protein content of wheat and maize was 7.9–15.1% and 5.6–12.6%. Organic manure treatments yielded significantly lower wheat grain protein content but had no significant effect on maize grain protein content relative to NPK treatment. Nitrogen uptake efficiency and nitrogen use efficiency at the cropping system level were 0.67–1.16 and 35.7–60.5 kg kg$^{-1}$. Phosphorus uptake efficiency and phosphorus use efficiency were 0.28–0.75 and 167–531 kg kg$^{-1}$. Compared with NPK treatment, 50% M, 75% M, and 100% M improved nitrogen use efficiency but decreased nitrogen uptake efficiency and phosphorus efficiencies. Annual N and P budgets were −1.3–79.1 kg ha$^{-1}$ a$^{-1}$ and 25.6–100.1 kg ha$^{-1}$a$^{-1}$, and both increased with the increase in organic manure input. Based on crop yield, grain protein content, nitrogen, and phosphorus efficiency and their budget, substitution of 25% inorganic fertilizer with organic manure is the rational combination under the winter wheat–summer maize system on an Anthrosol.

**Keywords:** wheat; maize; grain protein content; nitrogen efficiency; phosphorus efficiency

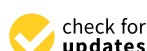



## 1. Introduction

Fertilization is a routine measure to ensure crop yield, but overfertilization remains a common problem in China. For example, with wheat, around 64% of farmers in the Fenhe Plain of Shanxi Province apply excessive N, and 78% apply excessive P relative to the recommended levels [1]. At the national scale, 75% of farmers use more fertilizers than the optimum for rice, wheat, and maize, which causes low N and P use efficiencies [2], increases soil N and P accumulation [3], and results in soil acidification and degradation [4,5], nitrate pollution of groundwater, eutrophication of surface waters, and greenhouse gas emissions [6,7]. To reduce these adverse effects, while maintaining crop yields and improving N and P efficiencies, the Chinese government released the zero-growth policy in fertilizer use in 2015 [8] and also highlighted greater recycling of agricultural byproducts, particularly organic manure. The total organic manure (ca. 2800 Mt in 2010, fresh weight) could provide about 16.4 Mt N and 5.3 Mt P in the country [9]. However, just 30–40% of

manure is currently recycled into croplands, while the rest is discarded and results in N and P pollution of the atmosphere and water [9–11]. Thus, making full use of organic manure can achieve multiple goals, including mitigating non-point source pollution [12], improving soil fertility, increasing crop yield, improving grain quality [13–15], and achieving the zero-growth policy in fertilizer use.

Many studies have demonstrated that the combination of organic manure and mineral fertilizer is best for achieving high crop yield [13,14,16]. Nevertheless, variable yield responses to the optimum combination of inorganic fertilizer with organic manure have been reported under different cropping systems and soils. For example, substituting 50% of inorganic N with organic manure maintained grain yield in the intensive maize–wheat, double cropping system on a silty loam [17] or obtained similar or increased maize yield on a silty-clay loam relative to NPK alone [18]. Long-term experiments (>30 years) on red clay soil have revealed that substitution of inorganic N with 30–70% with organic N increases crop yield in a double rice cropping system, and substitution of 70% inorganic N has shown the best performance [19,20]. In contrast, seven long-term experiments (12–15 years) found that replacement of inorganic fertilizer with organic manure significantly reduced rice yield in the rice–wheat system on sandy loam, silt loam, and clay soils [21]. These results reflect that the sound combination of organic manure with inorganic fertilizer requires further investigation for a given soil type and cropping system.

Application of organic manure impacts nutrient efficiency, but the results are inconsistent. A meta-analysis found that application of organic manure increased N and P uptake of rice, thus improving both N and P use efficiency (grain yield/nutrient uptake) [22]. Others have documented that organic manure combined with inorganic fertilizer reduced both N and P uptake efficiency (nutrient uptake/nutrient input) under the double rice system [23] and reduced P uptake efficiency compared with inorganic fertilizer alone under both wheat–fallow and wheat–maize systems [24]. Analyzing 15-year field experiments under the wheat–maize double cropping system at four sites in China, Duan et al. [25] reported that substituting 70% of inorganic N with organic manure maintained or improved N uptake efficiency relative to inorganic fertilizer alone. The differences among studies may be related to cropping system, soil type, and climate conditions that affect mineralization of organic manure and subsequently nutrient supply and nutrient uptake, etc. [22,24,25].

The Anthrosol, derived from loess material, is one of the major soil types in the Guanzhong Plain of northern China. It covers an area of up to $97.6 \times 10^4$ ha [26] and accounts for 34% of the arable land in Shaanxi Province [27]. The winter wheat–summer maize double cropping system is the main cropping system cultivated on Anthrosol and produces more than 70% of wheat and maize for Shaanxi Province. Previous studies have evaluated crop yield and N and P uptake efficiencies, under substituting 70% of inorganic N with organic manure in long-term experiments [24,25], as well as crop yield and nitrous oxide emission under various substitution proportions of inorganic fertilizer with organic manure through the combination of short-term experiments with long-term simulation [28]. However, there is a lack of information about the dynamics of crop yield, grain protein, and N and P efficiencies under various substitution proportions of inorganic fertilizer with organic manure under this cropping system. This study, therefore, investigated crop yield, grain protein content, N and P efficiencies, and N and P balance in response to various substitution ratios of inorganic fertilizer with organic manure in a 6-Year field experiment. Our results provide a reference for organic substitution of inorganic fertilizer to achieve high yield and greater efficiency and low environmental footprint for similar soil and cropping systems.

## 2. Materials and Methods

### 2.1. Experimental Site

A field experiment was conducted in Yangling (34°17′51″ N, 108°00′48″ E, 534 m asl), Guanzhong Plain, Shaanxi Province, China, during 2014–2020, where a winter wheat–summer maize (*Triticum aestivum* L., *Zea mays* L.), double cropping system was imple-



mented. This region displays a typical warm temperate continental monsoon climate, with a mean annual temperature of 12.9 °C, annual precipitation of 550–600 mm, and annual evaporation of 993 mm. Approximately 60% of precipitation occurs from July to September. The soil at the site is a silty-clay loam soil (clay 16%, silt 52%, and sand 32%), derived from loess material, and is classified as Eumorthic Anthrosol [29]. At the beginning of this experiment, the general soil properties in the topsoil layer (0–20 cm) were as follows: bulk density of 1.34 g cm$^{-3}$, pH (1:1 soil: water) approximately 8.6, soil organic matter content of 15.1 g kg$^{-1}$, total nitrogen content 0.68 g kg$^{-1}$, available phosphorus 10.1 mg kg$^{-1}$, and available potassium 165.6 mg kg$^{-1}$.

*2.2. Experimental Design*

The experiment was executed on 30 m$^2$ (7.5 m × 4 m) plots in triplicate using a randomized complete block design. Five treatments were applied in this study: (1) 100% of inorganic N, P and K fertilizers (NPK), (2) 75% NPK plus 25% organic manure (25% M), (3) 50% NPK plus 50% organic manure (50% M), (4) 25% NPK plus 75% organic manure (75% M), and (5) 100% organic manure (100% M). The nutrient application rates on NPK treatment were locally recommended for the cropping system. The detailed nutrient input for treatments is given in Table 1. Nitrogen fertilizer was applied in the form of urea before sowing wheat and at 6 leaf stages in the maize season. Organic manure and P and K fertilizers were applied in the form of dairy manure, calcium superphosphate, and potassium sulfate before wheat sowing. The mean contents of organic C, N, P, and K in the manure were 347.2 g kg$^{-1}$, 12.1 g kg$^{-1}$, 7.9 g kg$^{-1}$, and 11.8 g kg$^{-1}$, respectively, from 2014 to 2019, and manure rate was based on its N concentration. Thus, P and K inputs were different between treatments (Table 1).

**Table 1.** Details of fertilizer treatments and fertilizer rates for winter wheat–summer maize cropping system (kg ha$^{-1}$).

| Treatments | Winter Wheat | | | Summer Maize | | | Total | | |
|---|---|---|---|---|---|---|---|---|---|
| | N | P | K | N | P | K | N | P | K |
| NPK | 165 + 0 | 65 + 0 | 75 + 0 | 180 | 0 | 0 | 345 | 65 | 75 |
| 25% M | 124 + 86 | 49 + 38 | 56 + 25 | 135 | 0 | 0 | 345 | 87 | 81 |
| 50% M | 83 + 172 | 33 + 76 | 38 + 50 | 90 | 0 | 0 | 345 | 109 | 88 |
| 75% M | 41 + 259 | 16 + 114 | 19 + 75 | 45 | 0 | 0 | 345 | 131 | 94 |
| 100% M | 0 + 345 | 0 + 153 | 0 + 100 | 0 | 0 | 0 | 345 | 153 | 100 |

Note: The number after "+" represents the amount of N, P and K from organic manure.

The varieties for wheat and maize were Xiaoyan 22 and Zhengdan 958. The rainfall and temperature during the experiment period (2014–2020) are shown in Figure 1. Irrigation was conducted 0–1 times in the wheat season and 0–4 times in the maize season. Depending on the pattern of precipitation, each irrigation was about 80 mm. The vegetative stages of maize in 2017 and the whole growth period of wheat in 2019 were very dry; additionally, wheat in spring in 2018 suffered from frost. The chemicals were applied to control weeds, diseases, and insects based on recommendations.

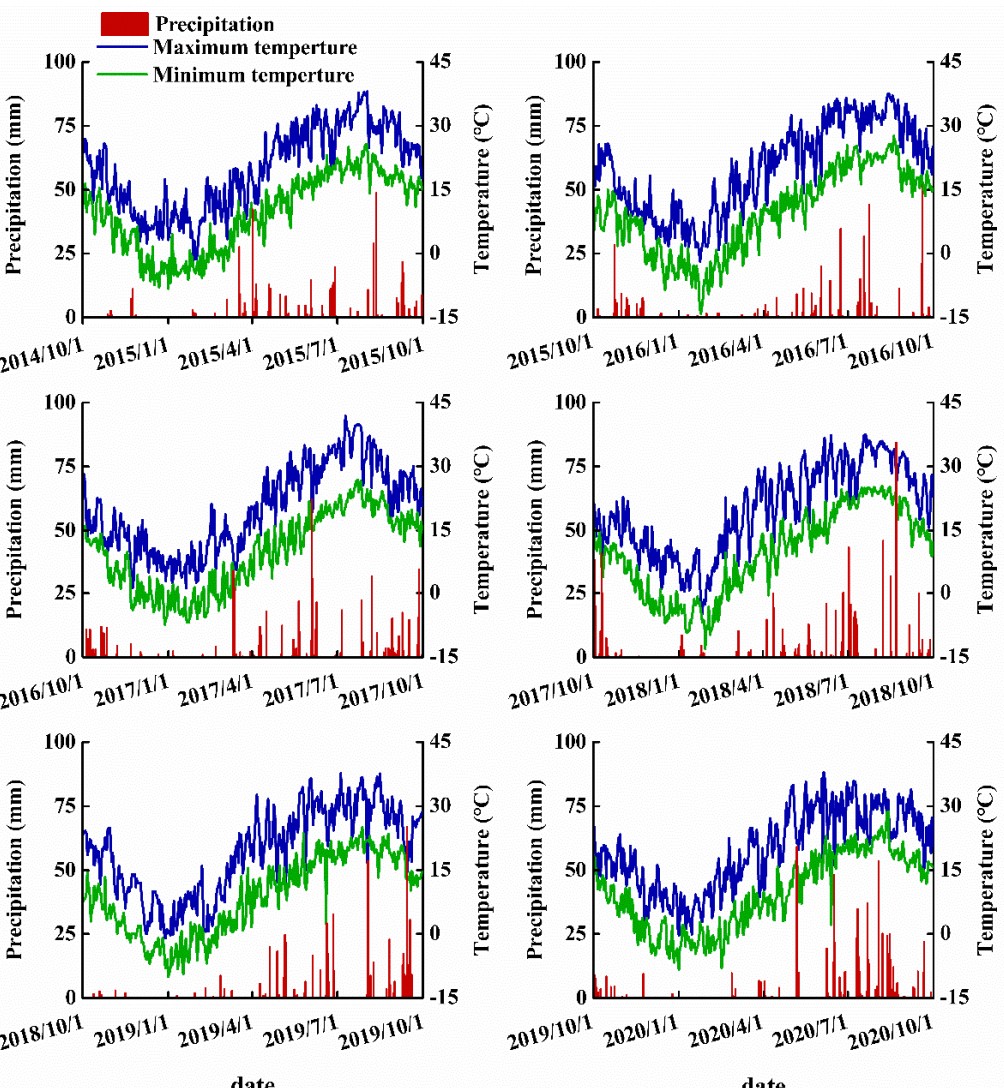

**Figure 1.** Daily maximum and minimum temperatures and precipitation during 2014–2020. Source: National Meteorological Observing Station in Yangling, Shaanxi Province, China.

### 2.3. Sampling and Measurements

Wheat and maize were sampled at maturity every year (sample area was 8 m$^2$ for wheat and 20 m$^2$ for maize) to determine grain yield and aboveground biomass. Subsamples of grain and straw were dried and milled to analyze N and P concentrations. Nitrogen content was determined following the method reported by Lv et al. [30], and phosphorus content was determined by the molybdenum blue colorimetric method [31].

### 2.4. Calculation and Statistical Analysis

Grain protein content (GPC, %) was calculated as follows [32]:

$$GPC\ (\%) = N_{grain}\ (\%) \times F_{N-Pro} \tag{1}$$

where $N_{grain}$ is the N concentration of grain, $F_{N-Pro}$ is the nitrogen-to-protein conversion factor, and wheat and maize are 5.70 and 6.25, respectively [32].

Nitrogen (phosphorus) uptake efficiency (N(P)PE) and nitrogen (phosphorus) use efficiency (N(P)UE, kg kg$^{-1}$) were calculated as follows [33,34]:

$$N(P)PE = (U_{WN(P)} + U_{MN(P)})\ (kg\ ha^{-1})/A_{N(P)}\ (kg\ ha^{-1}) \tag{2}$$

$$N(P)UE \ (kg \ kg^{-1}) = (Y_W + Y_M) \ (kg \ ha^{-1})/(U_{WN(P)} + U_{MN(P)}) \ (kg \ ha^{-1}) \qquad (3)$$

where $U_{WN(P)}$ and $U_{MN(P)}$, indicate N(P) uptake of wheat and maize, respectively; $A_{N(P)}$ is the annual N(P) application rate; and $Y_W$ and $Y_M$ are grain yields of wheat and maize.

N or P budget $(N(P)_b$, kg ha$^{-1}$) was calculated as the difference between total N(P) input $(F_{N(P)}$, kg ha$^{-1}$) and N(P) uptake by the crop $(U_{N(P)}$, kg ha$^{-1}$), as described by Khan et al. [24].

$$N(P)_b = F_{N(P)} - U_{N(P)} \qquad (4)$$

The effects of season, treatment, and their interactions on crop yield, grain protein content, nitrogen uptake and use efficiency, and phosphorus uptake and use efficiency were tested using repeated measures ANOVA (general linear model). When F-values were significant, multiple comparisons of means were performed using the least significant difference method (LSD) at 0.05 probability. All statistical analysis was performed with SPSS 18.0.

## 3. Results

### 3.1. Crop Yield

Wheat yield varied from 1643 to 8438 kg ha$^{-1}$ during six experimental seasons (Figure 2). ANOVA indicated significant seasonal effects on wheat yield as follows: 2018 < 2019 < 2020 < 2015 and 2016. Compared with 25% M, 100% M significantly reduced wheat yield, while other treatments showed similar yield to the above two treatments. Season significantly interacted with treatment impacting wheat yield; for example, both 25% M and 50% M treatments showed similar or higher yield relative to NPK treatment in 2016, 2017, 2018, and 2019, but the opposite was true in other seasons.

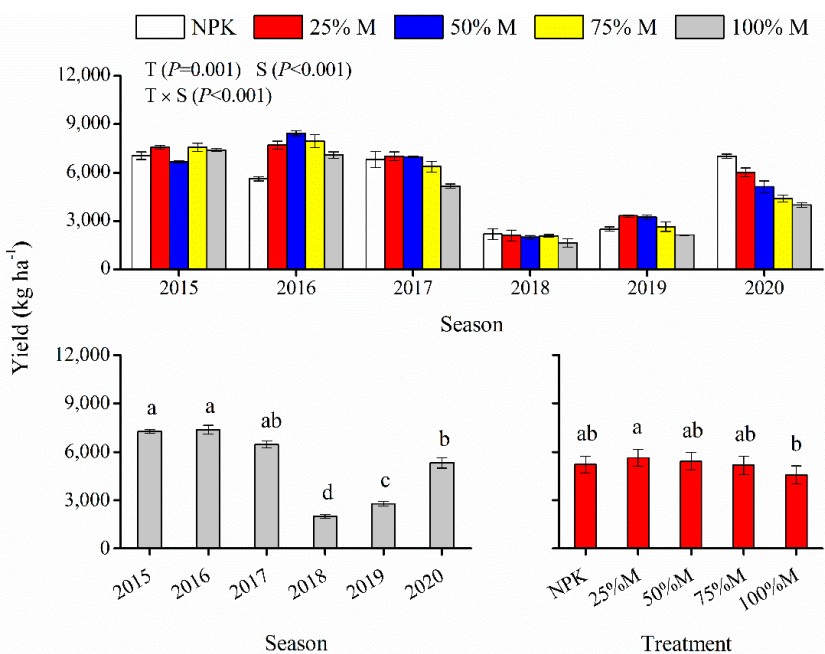

**Figure 2.** Wheat grain yield under different treatments of winter wheat–summer maize cropping system during 2014–2020. Bars represent standard error. Different letters above bars are significantly different between treatments or years at *P* < 0.05.

Maize yield ranged from 4847 to 11,104 kg ha$^{-1}$ (Figure 3). Maize yield in 2018 was significantly higher than that in the remaining seasons, and it was also higher in 2016, 2019, and 2020 than in 2015 and 2017. Additionally, season and treatment presented a significant interactive effect on maize yield. For example, organic manure treatments showed distinctly lower yield than NPK treatment in 2016 and 2017, but no difference was observed in the other four seasons.

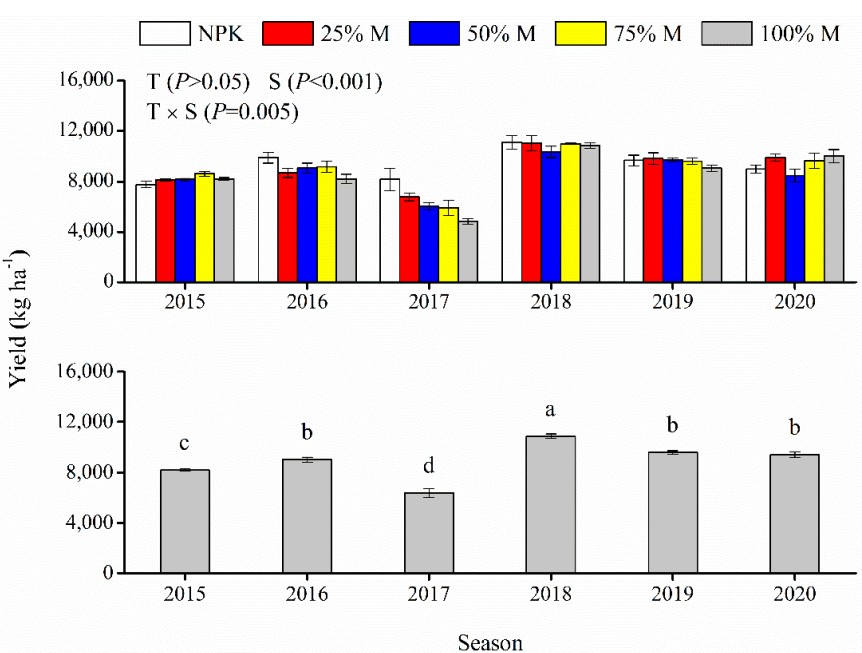

**Figure 3.** Maize grain yield under different treatments of winter wheat–summer maize cropping system during 2014–2020. Bars represent standard error. Different letters above bars are significantly different between treatments or years at *P* < 0.05.

Annual yield (wheat plus maize) ranged from 10,007 to 17,496 kg ha$^{-1}$ (Figure 4). Season, treatment, and their interaction affected annual yield. It was significantly lower in 2017, 2018, and 2019 than in the remaining seasons, and the annual yield in 2020 was also lower than that in 2016. Both NPK and 25% M treatments presented significantly greater annual yield compared to 100% M treatment. In 2015, 2016, and 2019, 25% M, 50% M, and 75% M treatments showed higher crop yield relative to NPK treatment, whereas lower values were observed in other seasons, especially in 2017.

### 3.2. Grain Protein Content

Grain protein content in wheat ranged from 7.9% to 15.1% (Figure 5). Compared with NPK, substitution of inorganic fertilizer with organic manure significantly decreased wheat grain protein content by 9.7–16.4%. Grain protein content was significantly higher in 2019 than in the other seasons. Season and treatment interaction affected grain protein content; for example, all treatments presented similar grain protein content in 2018, but organic manure treatments generally gave lower values in other seasons.

Grain protein content in maize ranged from 5.6% to 12.6% (Figure 6). Season affected maize grain protein content with a significantly higher value in 2017 than in other seasons.

### 3.3. Nitrogen and Phosphorus Efficiency

The NPE of the cropping system ranged from 0.67 to 1.16 (Table 2). Treatment and season significantly affected NPE. Compared with NPK, 50% M, 75% M and 100% M treatments reduced NPE by 13.0%, 15.0%, and 23.0%, respectively. The NPE was significantly higher in 2014–2015 and 2015–2016 than in other seasons (except for 2016–2017). Treatment and season interaction affected NPE, for example, 50% M, 75% M, and 100% M showed similar NPE values relative to NPK treatment in 2014–2015 and 2015–2016, while lower values were observed in other seasons.

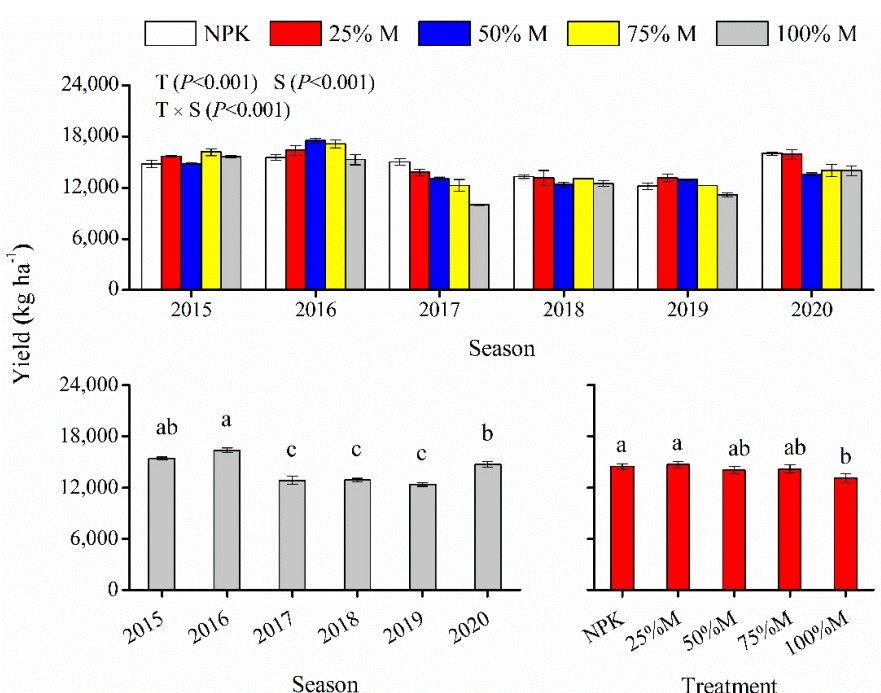

**Figure 4.** Annual grain yield under different treatments of winter wheat–summer maize cropping system during 2014–2020. Bars represent standard error. Different letters above bars are significantly different between treatments or years at *P* < 0.05.

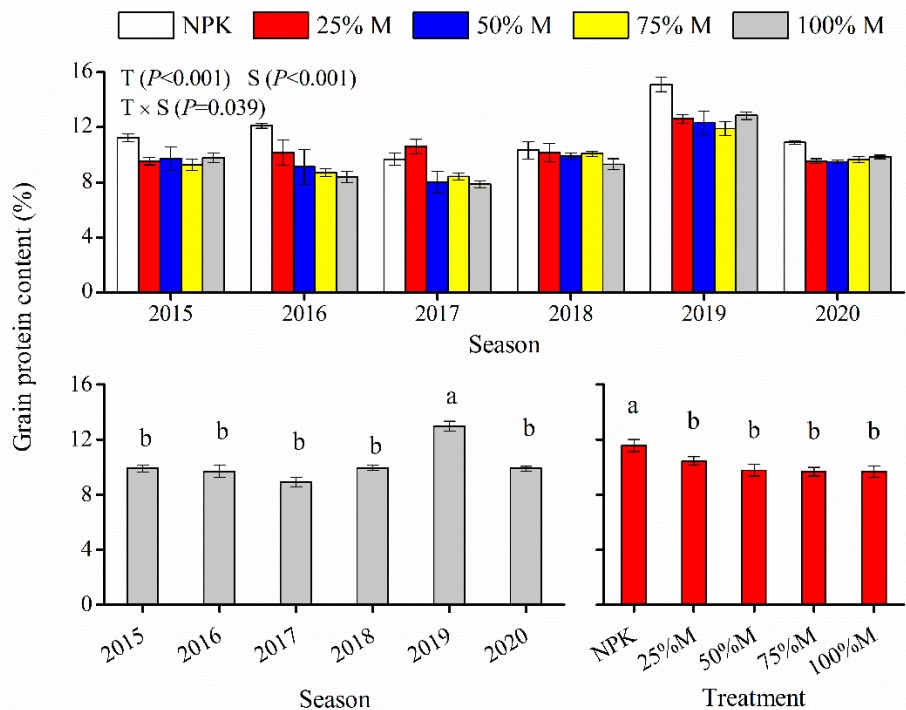

**Figure 5.** Wheat grain protein content under different treatments of winter wheat–summer maize cropping system during 2014–2020. Bars represent standard error. Different letters above bars are significantly different between treatments or years at *P* < 0.05.

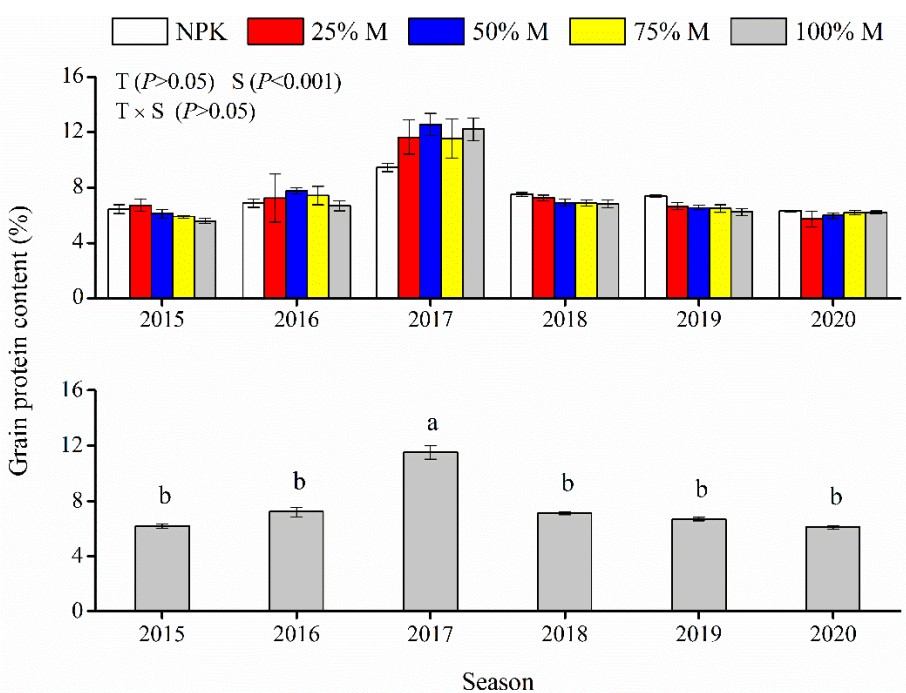

**Figure 6.** Maize grain protein content under different treatments of winter wheat–summer maize cropping system during 2014–2020. Bars represent standard error. Different letters above bars are significantly different between treatments or years at $P < 0.05$.

**Table 2.** Effect of year and treatment on nitrogen uptake efficiency (NPE), nitrogen use efficiency (NUE), phosphorus uptake efficiency (PPE) and phosphorus use efficiency (PUE), of winter wheat–summer maize cropping system during 2014–2020.

| Season | Treatment | NPE | NUE | PPE | PUE |
|---|---|---|---|---|---|
| | | (kg kg$^{-1}$) | | | |
| 2014–2015 | NPK | 1.07 ± 0.07 a | 40.46 ± 2.29 b | 0.56 ± 0.02 a | 405.35 ± 8.31 a |
| | 25% M | 0.98 ± 0.03 a | 46.38 ± 1.28 a | 0.45 ± 0.03 b | 401.54 ± 25.51 a |
| | 50% M | 0.94 ± 0.06 a | 45.93 ± 2.80 ab | 0.37 ± 0.04 bc | 377.51 ± 43.86 a |
| | 75% M | 0.97 ± 0.04 a | 48.28 ± 0.67 a | 0.37 ± 0.02 c | 340.61 ± 6.37 a |
| | 100% M | 0.90 ± 0.02 a | 50.31 ± 1.37 a | 0.32 ± 0.01 c | 327.07 ± 10.81 a |
| 2015–2016 | NPK | 1.03 ± 0.07 ab | 44.14 ± 2.48 a | 0.45 ± 0.01 a | 530.61 ± 6.09 a |
| | 25% M | 1.11 ± 0.02 a | 42.80 ± 2.13 a | 0.39 ± 0.02 b | 470.71 ± 3.96 b |
| | 50% M | 1.16 ± 0.03 a | 43.99 ± 1.91 a | 0.36 ± 0.01 c | 437.97 ± 8.87 c |
| | 75% M | 1.06 ± 0.08 a | 47.27 ± 2.40 a | 0.32 ± 0.00 d | 398.09 ± 8.91 d |
| | 100% M | 0.87 ± 0.03 b | 50.93 ± 1.90 a | 0.28 ± 0.00 e | 340.09 ± 13.79 e |
| 2016–2017 | NPK | 1.11 ± 0.05 a | 39.37 ± 0.95 a | 0.53 ± 0.02 b | 433.76 ± 7.46 a |
| | 25% M | 1.11 ± 0.07 a | 36.17 ± 2.27 a | 0.56 ± 0.03 b | 381.33 ± 18.52 a |
| | 50% M | 0.89 ± 0.01 b | 42.23 ± 0.90 a | 0.54 ± 0.04 b | 377.26 ± 20.85 a |
| | 75% M | 0.84 ± 0.03 b | 42.54 ± 2.11 a | 0.65 ± 0.03 a | 293.66 ± 23.51 b |
| | 100% M | 0.68 ± 0.02 c | 42.62 ± 1.66 a | 0.56 ± 0.04 b | 281.29 ± 22.29 b |
| 2017–2018 | NPK | 0.92 ± 0.01 a | 42.08 ± 0.73 a | 0.72 ± 0.01 a | 284.96 ± 10.36 a |
| | 25% M | 0.88 ± 0.05 ab | 43.21 ± 1.36 a | 0.51 ± 0.03 b | 261.11 ± 3.84 ab |
| | 50% M | 0.73 ± 0.04 c | 49.04 ± 2.05 a | 0.36 ± 0.01 c | 259.83 ± 11.8 ab |
| | 75% M | 0.80 ± 0.01 bc | 47.53 ± 0.33 a | 0.32 ± 0.02 cd | 251.55 ± 14.58 b |
| | 100% M | 0.79 ± 0.04 bc | 46.53 ± 3.75 a | 0.29 ± 0.00 d | 213.74 ± 7.16 c |

Note: different lowercase letters in the same column represent significant difference between treatments in the same season or between seasons across treatments or treatments across seasons ($P < 0.05$).

**Table 2.** *Cont.*

| Season | Treatment | NPE | NUE | PPE | PUE |
|---|---|---|---|---|---|
| | | (kg kg$^{-1}$) | | | |
| 2018–2019 | NPK | 0.99 ± 0.04 a | 35.74 ± 1.36 b | 0.62 ± 0.03 a | 301.14 ± 4.13 a |
| | 25% M | 0.85 ± 0.05 b | 44.97 ± 1.29 a | 0.49 ± 0.01 b | 283.97 ± 1.95 a |
| | 50% M | 0.81 ± 0.01 bc | 46.24 ± 0.71 a | 0.42 ± 0.01 bc | 250.28 ± 6.56 a |
| | 75% M | 0.76 ± 0.06 bc | 47.01 ± 3.21 a | 0.38 ± 0.04 c | 213.04 ± 21.05 ab |
| | 100% M | 0.71 ± 0.01 c | 45.68 ± 1.36 a | 0.37 ± 0.02 c | 167.46 ± 5.40 b |
| 2019–2020 | NPK | 0.91 ± 0.01 a | 50.80 ± 0.94 b | 0.75 ± 0.01 a | 326.83 ± 7.20 a |
| | 25% M | 0.78 ± 0.02 b | 58.83 ± 1.71 a | 0.61 ± 0.03 b | 294.65 ± 2.94 b |
| | 50% M | 0.69 ± 0.04 c | 57.20 ± 2.78 a | 0.47 ± 0.01 c | 259.32 ± 3.93 c |
| | 75% M | 0.70 ± 0.02 bc | 58.06 ± 2.22 a | 0.43 ± 0.02 cd | 241.65 ± 4.19 d |
| | 100% M | 0.67 ± 0.04 c | 60.48 ± 2.93 a | 0.38 ± 0.01 d | 231.46 ± 5.24 d |
| Season | 2014–2015 | 0.97 ± 0.02 a | 46.27 ± 1.12 b | 0.23 ± 0.02 bc | 679.51 ± 28.27 ab |
| | 2015–2016 | 1.04 ± 0.03 a | 45.83 ± 1.14 b | 0.36 ± 0.02 c | 435.49 ± 17.56 a |
| | 2016–2017 | 0.93 ± 0.05 ab | 40.59 ± 0.92 c | 0.57 ± 0.02 a | 353.46 ± 17.05 b |
| | 2017–2018 | 0.82 ± 0.02 b | 45.68 ± 1.04 b | 0.44 ± 0.04 bc | 254.24 ± 7.29 c |
| | 2018–2019 | 0.82 ± 0.03 b | 43.93 ± 1.30 b | 0.46 ± 0.03 bc | 243.18 ± 13.52 c |
| | 2019–2020 | 0.75 ± 0.03 b | 57.08 ± 1.23 a | 0.53 ± 0.04 ab | 270.78 ± 9.62 c |
| Treatment | NPK | 1.00 ± 0.02 a | 42.10 ± 1.26 b | 0.57 ± 0.04 a | 429.53 ± 37.72 a |
| | 25% M | 0.95 ± 0.03 ab | 45.39 ± 1.75 ab | 0.47 ± 0.03 b | 407.42 ± 41.56 ab |
| | 50% M | 0.87 ± 0.04 bc | 47.44 ± 1.37 a | 0.39 ± 0.03 c | 371.84 ± 37.44 bc |
| | 75% M | 0.85 ± 0.03 cd | 48.45 ± 1.34 a | 0.38 ± 0.04 c | 345.93 ± 39.38 cd |
| | 100% M | 0.77 ± 0.02 d | 49.42 ± 1.59 a | 0.34 ± 0.03 c | 309.18 ± 37.26 d |
| Source | | | | | |
| Season (S) | | <0.001 | <0.001 | <0.001 | <0.001 |
| Treatment (T) | | <0.001 | <0.001 | <0.001 | <0.001 |
| S × T | | 0.001 | 0.279 | <0.001 | 0.063 |

Note: different lowercase letters in the same column represent significant difference between treatments in the same season or between seasons across treatments or treatments across seasons ($P < 0.05$).

The NUE of the cropping system ranged from 35.7 to 60.5 kg kg$^{-1}$ (Table 2). Season and treatment significantly affected NUE. Compared with NPK, 50% M, 75% M and 100% M increased NUE by 12.7%, 15.1% and 17.4%, respectively. The NUE was highest in 2019–2020 and lowest in 2016–2017.

The range of PPE was 0.28 to 0.75 (Table 2). Compared with NPK, organic manure treatments decreased PPE by 18.0–39.3%. The PPE was lower in 2015–2016 than in the 2016–2017 and 2019–2020 seasons. Additionally, 25% M, 50% M, 75% M and 100% M treatments showed similar or higher PPE values than NPK treatment in 2016–2017, but the opposite was true in the other seasons.

The range of PUE was 167 to 531 kg kg$^{-1}$ (Table 2). Relative to NPK treatment, 50% M, 75% M, and 100% M treatments remarkably decreased PUE by 14.0%, 23.8%, and 31.6%, respectively. The PUE showed lower values in 2017–2018, 2018–2019, and 2019–2020 seasons than in the remaining seasons.

*3.4. Nitrogen and Phosphorus Budget*

Nitrogen budget was from −1.3 kg ha$^{-1}$ a$^{-1}$ on NPK treatment to 79.1 kg ha$^{-1}$ a$^{-1}$ on 100% M treatment (Table 3). Phosphorus budget was from 25.6 to 100.1 kg ha$^{-1}$ a$^{-1}$, the lowest on NPK, and the highest on 100% M treatment (Table 4).

**Table 3.** Apparent nitrogen balance of winter wheat–summer maize cropping system under different fertilization treatments during 2014–2020.

| Item (kg ha$^{-1}$) | | NPK | 25% M | 50% M | 75% M | 100% M |
|---|---|---|---|---|---|---|
| N input | Manure N | 0 | 517 | 1035 | 1552 | 2070 |
| | Fertilizer N | 2070 | 1553 | 1035 | 518 | 0 |
| N output | 2014–2015 | 368.8 | 338.6 | 325.1 | 335.2 | 311.0 |
| | 2015–2016 | 354.2 | 383.3 | 398.7 | 365.5 | 300.5 |
| | 2016–2017 | 381.8 | 384.0 | 308.1 | 288.5 | 235.4 |
| | 2017–2018 | 316.1 | 303.9 | 252.7 | 275.0 | 271.0 |
| | 2018–2019 | 341.7 | 292.8 | 280.3 | 263.3 | 244.5 |
| | 2019–2020 | 315.0 | 270.1 | 238.7 | 241.6 | 232.8 |
| | sum | 2077.7 | 1972.6 | 1803.7 | 1769.1 | 1595.2 |
| N budget | | −7.7 | 97.4 | 266.3 | 300.9 | 474.8 |
| Annual N budget | | −1.3 | 16.2 | 44.4 | 50.1 | 79.1 |

**Table 4.** Apparent phosphorus balance of winter wheat–summer maize cropping system under different fertilization treatments during 2014–2020.

| Item (kg ha$^{-1}$) | | NPK | 25% M | 50% M | 75% M | 100% M |
|---|---|---|---|---|---|---|
| P input | Manure P | 0 | 229 | 458 | 687 | 916 |
| | Fertilizer P | 390 | 293 | 195 | 98 | 0 |
| P output | 2014–2015 | 36.6 | 39.3 | 40.4 | 47.5 | 47.9 |
| | 2015–2016 | 29.2 | 34.8 | 40.0 | 43.1 | 45.0 |
| | 2016–2017 | 34.6 | 36.3 | 34.8 | 42.0 | 35.9 |
| | 2017–2018 | 46.7 | 50.3 | 47.6 | 52.3 | 58.5 |
| | 2018–2019 | 40.5 | 46.2 | 51.8 | 58.8 | 66.9 |
| | 2019–2020 | 49.0 | 54.0 | 52.4 | 58.0 | 60.6 |
| | sum | 236.6 | 261.0 | 267.0 | 301.8 | 314.8 |
| P budget | | 153.4 | 260.4 | 385.8 | 482.4 | 600.8 |
| Annual P budget | | 25.6 | 43.4 | 64.3 | 80.4 | 100.1 |

## 4. Discussion

### 4.1. Response of Crop Yield and Grain Protein Content to Organic Manure Substitution

The six-year experiment found that the 25% organic manure replacement had the best effect on crop yield on the Anthrosol tested. This is consistent with studies on similar soils, which showed that the substitution of 25% inorganic fertilizer with organic manure maintained crop yield under a wheat–maize cropping system while substituting 30–100% inorganic fertilizer with organic manure reduced crop yield [13,35]. Long-term experiments (14–22 years) also demonstrated that the replacement of 50% to 100% inorganic fertilizer with organic manure risked lower yields in the beginning years but maintained/increased crop yields in the later years under both wheat–fallow and wheat–maize rotations on alkaline soils [25,36,37]. On acidic soils, replacement of 50–100% inorganic fertilizer with organic manure significantly reduced wheat yield and maintained/increased rice yield under a wheat-rice rotation system based on 1–6 years of experiments [15,38]. Long-term experiments revealed that substituting 30–70% inorganic fertilizer with organic manure could maintain or increase crop yield under wheat–maize, wheat-rice, and double rice cropping systems, and 70% M treatment was the best [19,20,25,39]. A meta-analysis found that substitution of inorganic fertilizer less than 30% with organic manure could increase wheat yield, when substitution exceeds 30%, wheat yield increase can be detected after more than 10 years or on acidic soils [16]. Overall, the substitution of inorganic fertilizer with organic manure in a low proportion generally maintains or increases crop yield on alkaline soils, which could be ascribed to: (1) other mineral nutrients apart from N, P, and K added by organic manure [40,41]; (2) improvement of soil physical properties [36,42], which improve the supply and retention capacity of water and nutrients, and thus root penetration and nutrient and water uptake [43,44]; (3) improving soil enzyme activity [45,46] and soil

microbial diversity and community, thus increasing soil productivity [47,48]. The high proportion of organic manure substituting inorganic fertilizer generating yield penalty as observed in this study is possibly related to low available nutrient content caused by the slow mineralization rate of organic manure under suboptimal environmental conditions (e.g., soil hydrothermal status), thus, limiting crop growth [49,50].

Protein is an important part of grain quality. Substitution of inorganic fertilizer by manure decreased wheat grain protein content and had no effect on maize grain protein content compared with inorganic fertilizer alone. Similarly, rice grain protein content decreased following the increase in the proportion of organic manure application [51]. Wang et al. [52] documented that 30–100% biogas slurry replacing inorganic fertilizer did not affect wheat grain protein content. Others have found that the replacement of inorganic fertilizer with organic manure (25% M or 30% M) increased grain protein content compared with inorganic fertilizer alone [12,53]. The variability of the results among studies might be related to N supply and uptake. The proportion of N translocation at pre-anthesis accounted for 51–91% of grain N content in wheat [54,55]. The lack of available N supply in the early crop growth may affect the N uptake and translocation [51]. The organic manure with a C/N ratio >25 applied in this study might induce competition from microbial decomposers for available N. The availability of mineral N for crop growth might be lower in the manure treatments compared to NPK treatment [36], especially for winter wheat that grows at low temperature at the early stage, which could impact the mineralization of manure. Considering the protein content of grains, the proportion of organic manure replacing inorganic fertilizer should not be high, especially at the beginning of the experiment. With the improvement of soil fertility, nutrient supply capacity is enhanced, and the proportion of organic manure application can be gradually increased [36]. Foliar fertilization of N after anthesis might be a viable measure to increase grain protein concentration [56].

*4.2. Response of Nitrogen Efficiency to Organic Manure Substitution*

Nitrogen use efficiency refers to the proportion of nitrogen absorbed by the plants to be converted to grain yield [57]. Organic manure substitution exceeding 25% significantly increased the NUE of the cropping system compared to the sole NPK (Table 2), which was similar to results obtained under a single rice system on an Entisol [58]. Other short- and long-term studies (1–30 years) documented that the substitution of organic manure for inorganic fertilizer increased NUE in both single and double rice cropping systems [15,51,59,60]. Li et al. [61] documented that most organic manure substitution treatments presented similar NUE to inorganic fertilizer alone under the winter wheat–summer maize cropping system based on a one−year study on light loam soil. The variation among studies is probably associated with the soil and climate conditions, crop type, and experiment duration, which affect the mineralization of organic manure [36]. The possible reasons for the improvement/maintenance of NUE by substituting inorganic fertilizer with organic manure include (1) the availability of N in organic manure is low, especially the replacement ratio of organic manure is high, which reduces the N accumulation in stem and leaf during the vegetative growth stage [51,60,61]; (2) organic manure application improves post-anthesis N supply, promotes post-anthesis dry matter accumulation, compensates N deficiency pre-anthesis to some extent, and subsequently maintains crop yield [62,63], thus improving NUE.

Nitrogen uptake efficiency reflects the level of N management [64]. If NPE = 1, it means that the input nutrient can exactly meet the nutrient requirement of the crop; NPE < 1 indicates that N uptake is less than N input, and the unremoved N by the crop may store in the soil or enter into the environment; NPE > 1 means that the N removed is greater than the N input, indicating depletion of soil N, thereby a decrease in soil fertility. NPE in our study (0.77–1.00) was similar to the international recommended value of 0.5–0.9 [65], which was also the value reflected by the low N budget from −1.3 to 79.1 kg ha$^{-1}$ a$^{-1}$, less than the recommended 80 kg ha$^{-1}$ a$^{-1}$ [65]. These results highlight that the application

rate of inorganic N recommended in this experiment is appropriate under the cropping system.

The 25% M had similar NPE to the NPK while other manure treatments presented lower values. This was in agreement with Li et al. [61], who showed that 25% cow manure plus 75% inorganic N had similar NPE to the inorganic fertilizer alone, and 50%, 75%, and 100% cow manure substitution reduced NPE from one year experiment under same cropping system. Results from 15-year experiments on different soils indicated that substituting 70% of inorganic N with organic manure improved NPE in acidic soil and maintained NPE in other alkaline soils under a wheat–maize cropping system [25]. However, substitutions with pig manure or chicken manure with various proportions (25–100%) showed the same NPE as inorganic fertilizer alone under a winter wheat–summer maize cropping system on an alkaline soil [61]. Hou et al. [59] also reported that organic manure (applied as A. *sinicus* L. and swine) replacing 30%, 50%, and 70% of the inorganic fertilizer had no significant effect on NPE under a double rice cropping system on an acidic soil based on a 30-year field experiment. The inconsistent results might be related to the factors that influence the mineralization of organic manure and N uptake as discussed above and manure types with different properties [25,59,61].

### 4.3. Response of Phosphorus Efficiency to Organic Manure Substitution

The PUE was lower following the substitution of organic manure for inorganic fertilizer (Table 2). Khan et al. [24] also reported that compared with inorganic fertilizer alone, a combination of organic manure and inorganic fertilizer reduced PUE in both winter wheat–summer fallow and winter wheat–summer maize systems on the same soil. Similarly, Xin et al. [66] reported that both 50% and 100% compost substitution of inorganic fertilizer had significantly lower PUE under the wheat–maize system. Lu et al. [67] found that organic manure combined with inorganic fertilizer treatment increased PUE in the double rice cropping system. The reduction of PUE is partly due to the excessive P input caused by organic manure whose input has been based on N concentration in many experiments like the current study, leading to the excessive P input and luxury P uptake by crops (Table 4) [24]. The reason for the improvement of PUE is consistent with that of NUE because the slow mineralization rate of organic manure reduces crop P uptake at the early stage but increases P uptake at the late stage, thus ensuring crop yield and improving PUE [67].

Phosphorus uptake efficiency reflects the management of P. The PPE observed in the current study (0.37–0.61) was similar to the range of 0.20–0.54 reported in China [24], but lower than the value of 0.5–0.7 observed in England [34]. Low PPE in this study was also reflected by high values of P budget (25.6–100 kg ha$^{-1}$ a$^{-1}$), indicating the excessive application of P, especially in the treatment with organic manure. Previous studies have shown that each 100 kg P ha$^{-1}$ surplus would increase the Olsen-P by 2–6 mg P kg$^{-1}$ [68–70], which means long-term excessive application of P would eventually lead to a large quantity of P accumulation with high availability and thus increase potential environmental problems of P leaching and resource losses [24].

Substitution of inorganic fertilizer with organic manure reduced PPE (Table 2). On the same soil, Khan et al. [24] found that 70% organic manure substitution of inorganic N significantly decreased PPE under both wheat–maize and wheat–fallow (rain–fed) systems. Under the rice–lentil cropping system, Pradhan et al. [71] reported lower PPE was in treatment receiving inorganic fertilizer in combination with farmyard manure (50% farmyard manure) on neutral soil (pH ≈ 7). On the contrary, Lu et al. [67] reported that substituting inorganic fertilizer with pig manure has no effect on PPE under a double rice cropping system on acidic soil. Additionally, replacing inorganic fertilizer with pig manure or compost (mixture of wheat straw, oil rapeseed cake, and cottonseed cake) has been reported to increase PPE under a double rice cropping system on acidic soil [72] or wheat–maize double cropping system on an alkaline soil [66]. The difference among studies might mainly be associated with the amount of P input, environmental condition (e.g.,

soil and climate), crop type, and property of organic manure. Considering the limited P resources and potential risk of P release to the water body, organic manure substitution for fertilizers should be based on both N and P in the future.

## 5. Conclusions

The present study demonstrated that 75% inorganic fertilizer combined with 25% organic manure sustained crop yield, maintained maize grain protein content, and generally achieved similar nitrogen and phosphorous efficiency to the recommended inorganic fertilizers alone. Considering the reduction of wheat grain protein content, foliar nitrogen application might be an efficient measure to be adopted to improve grain protein content. Due to the high quantity of phosphorous left in soil, organic manure substitution for inorganic fertilizer should be simultaneously considered both nitrogen and phosphorous in the future to efficiently use organic resources and protect phosphate resources.

**Author Contributions:** Data analysis, Y.H., F.L., X.L., C.Z., X.Y. and S.Z.; data collection, Y.H., F.L., X.L. and C.Z.; figures, Y.H. and F.L.; funding acquisition, X.Y. and S.Z.; investigation, Y.H. and F.L.; study design, B.S., X.Y. and S.Z.; proofread the final version of the manuscript, B.S., X.Y. and S.Z.; writing—original draft, Y.H. and F.L.; writing—review and editing, B.S., X.Y. and S.Z. All authors have read and agreed to the published version of the manuscript.

**Funding:** This work was sponsored by the Ministry of Agriculture and Rural Affairs of China under Special funds for the Operation and Maintenance of Scientific Research Facilities (G2022-07-2) and the Natural Science Foundation of Shaanxi Province, China (2022JQ-288).

**Institutional Review Board Statement:** Not applicable.

**Informed Consent Statement:** Not applicable.

**Data Availability Statement:** Data is contained within the article.

**Acknowledgments:** We are incredibly grateful to all the staff involved in the running and managing of the experiment. The authors also greatly acknowledge Victor Sadras for his comments and language check, who is supported by the Foreign Expert Introduction Program sponsored by the Ministry of Science and Technology (F2011322005).

**Conflicts of Interest:** The authors declare no conflict of interest.

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
