# Peer review of "Crop Yield and Nutrient Efficiency under Organic Manure Substitution Fertilizer in a Double Cropping System: A 6-Year Field Experiment on an Anthrosol"

_agronomy, doi:10.3390/agronomy12092047_

Round 1

Reviewer 1 Report

Comments are marked in the manuscript.  The text of the figure is not always correct - see remark in manuscript. Occasional discrepancy between  significance.

Author Response

We have not received the comments from reviewer 1.

Reviewer 2 Report

Dear Authors

Interesting piece of research, I suggested the Authors to do the following:

please specify in the title 6 years filed study

please make a focus on Anthrosols in the introduction, what is their origin, why they play such a role in the region, why you need to amend them and why you suppose that the mineral + organic is the best solution.

The manuscript is concise but a bit of detail in the introduction is needed to put the reader into context, the same is for the discussions.

please dedicate the right attention in organizing the narration, start from the big picture to the field details. 

I would be very happy to see the revised version.

Please carefully check for typos.

Improve the quality of the figures.

Kind regards

Calogero Schillaci
